# A-NeRF: Articulated Neural Radiance Fields for Learning Human Shape, Appearance, and Pose

**Shih-Yang Su**[1]      **Frank Yu**[1]      **Michael Zollhöfer**[2]      **Helge Rhodin**[1]

[1]University of British Columbia          [2]Facebook Reality Labs

## Abstract

While deep learning reshaped the classical motion capture pipeline with feed-forward networks, generative models are required to recover fine alignment via iterative refinement. Unfortunately, the existing models are usually hand-crafted or learned in controlled conditions, only applicable to limited domains. We propose a method to learn a generative neural body model from unlabelled monocular videos by extending Neural Radiance Fields (NeRFs). We equip them with a skeleton to apply to time-varying and articulated motion. A key insight is that implicit models require the inverse of the forward kinematics used in explicit surface models. Our reparameterization defines spatial latent variables relative to the pose of body parts and thereby overcomes ill-posed inverse operations with an overparameterization. This enables learning volumetric body shape and appearance from scratch while jointly refining the articulated pose; all without ground truth labels for appearance, pose, or 3D shape on the input videos. When used for novel-view-synthesis and motion capture, our neural model improves accuracy on diverse datasets. Project website: https://lemonatsu.github.io/anerf/.

## 1   Introduction

Generative models have evolved from Generative Adversarial Networks (GANs) recreating images [14, 21] to neural scene representations [39, 56, 57] providing control and image understanding for downstream tasks via structured latent variables. However, most 3D models require 3D labels that cannot be crowd-sourced on natural images and require dedicated depth sensors. It is hence an important research problem to learn 3D representations from 2D observations, which is particularly challenging for humans with diverse body shapes and appearances and their non-rigid motion.

Modern human motion capture techniques typically combine the advantages of discriminative and generative approaches. A feed-forward 3D human pose estimation approach provides a rough initial estimate of human pose. Afterward, a generative approach based on either a high-quality 3D scan of the person [17], or a parametric human body model learned from laser scans [4] refines the estimate iteratively based on the image evidence. Although achieving unprecedented accuracy, existing models require a low-dimensional, restrictive, shape body model or a personalized 3D scan of the user.

We introduce *Articulated Neural Radiance Fields* (A-NeRF) for learning a user-specific neural 3D body model and underlying skeleton pose from unlabelled videos (see Figure 1). When applied to motion capture, it alleviates the need for template models while maintaining the advantages and accuracy of current generative approaches. A-NeRF extends Neural Radiance Fields (NeRF) [39] to work with single videos and articulated motion. NeRF parameterizes the scene implicitly as

$$F_\phi(\Gamma(\mathbf{q}), \Gamma(\mathbf{d})) \mapsto (\sigma, \mathbf{c}), \quad \text{with } \sigma \in \mathbb{R}, \mathbf{c} \in \mathbb{R}^3, \mathbf{q} \in \mathbb{R}^3, \text{ and } \mathbf{d} \in \mathbb{R}^3, \tag{1}$$

by chaining $F_\phi$, a Multi-layer Perceptron (MLP), with $\Gamma$, the Positional Encoding (PE) [66]. First, the PE maps the input scene point $\mathbf{q}$ and view direction $\mathbf{d}$ to a higher dimensional space that enables the

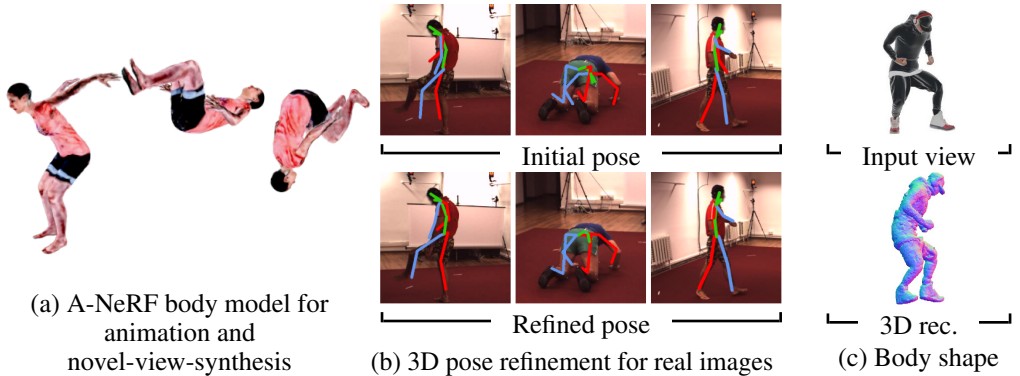

(a) A-NeRF body model for animation and novel-view-synthesis

(b) 3D pose refinement for real images

Initial pose

Refined pose

Input view

3D rec.

(c) Body shape

Figure 1: Our A-NeRF jointly learns a neural body model of the user and works with diverse body poses (left) while also refining the initial 3D articulated skeleton pose estimate from a single or, if available, multiple views without tedious camera calibration (center). Underlying is a template-free neural representation (right) and skeleton-based embedding coupled with volume volumetric rendering. ***Real faces and their reconstructions are blurred in all figures for anonymity.***

MLP to learn a meaningful scene representation function $F_\phi$ that subsequently outputs the radiance $\mathbf{c}$ and opacity $\sigma$ at every point in space. Second, the implicitly described scene (via conditioning on query locations) is rendered via classical ray-marching from computer graphics. The advantage of the MLP representation is that it avoids the complexity of volumetric grids [29], circumvents the artifacts induced by the implicit bias of screen-space convolution [40, 54], and, unlike surface meshes, can have flexible topology. However, the original NeRF only works for static scenes captured from dozens of calibrated cameras such that each 3D point is seen from multiple views.

Our conceptual contribution lies in learning a neural latent representation relative to an articulated skeleton. While explicit models such as the popular SMPL body model [30] deform a surface via forwards kinematics, the implicit form of A-NeRF makes us re-think how skeletons can be integrated—implicit networks require the inverse transformation from 3D world coordinates to the reference skeleton, a significantly harder task that has not been fully explored. Our core technical novelty is to come up with and evaluate different parameterizations of $\Gamma(\mathbf{q}), \Gamma(\mathbf{d})$ in Eq. 1 to create local coordinates relative to the articulated skeleton. Since a point in 3D world coordinates cannot uniquely be associated with a body part, we resolve the mentioned ill-posed inverse problem by overparameterizing with one embedding per bone. This embeds domain knowledge of how humans move and provides a common frame for the neural network to combine body shape and appearance constraints across the entire captured sequence (see Figure 2).

We demonstrate that all our contributions together enable learning of a neural body model from monocular video, requiring only rough 3D pose estimates for initialization, that reaches a level of detail previously only attained with parametric surface models or multi-view approaches [51].

**Scope.** We apply the model to motion capture, character animation, and appearance and motion transfer and demonstrate that the pose refinement improves on existing monocular skeleton reconstruction. We learn in the transductive setting, for a specific target video that is known at training time but has no ground truth. A-NeRF enables novel view synthesis of dynamic motions, with plausible however non-physical illumination. Additional steps are needed to enable relighting applications.

**General impact.** Building a self-supervised approach for personalized human body modelling promises to be more inclusive to people and activities that are not well represented in supervised datasets. However, it bears the risk that 3D models of people are created without consent. We urge users to only use datasets collected for developing and validating motion capture algorithms.

## 2 Related Work

Our approach builds upon and is related to the following work on human pose and shape estimation, human modeling, and neural scene representations [62].

**Discriminative Human Pose Estimation.**   While feed-forward estimation of the 3D joint positions [26, 32, 34, 35, 43, 44, 52, 59, 64, 69, 72] or joint angles and bone lengths [53, 73] of the skeleton is highly accurate, such discriminative estimates are prone to misalignment when overlayed onto the input image due to the generalization gap. The skeleton pose can be refined to better match the 2D pose estimates, but this usually leads to larger errors in 3D [36, 37]. We use [23] for initializing skeleton pose and combine it with a neural body model.

**Surface-based Generative Body Models.**   These are obtained by either constraining template meshes via deformation energies [17, 70] or learning parametric human body models from a large collection of laser scans [6, 9, 30]. Their low-dimensional parameters constrain the space of plausible human shapes and motions. This enables real-time reconstructions from single images [7, 15], detailed texturing and displacement mapping [3, 5], and alleviates manual rigging [2]. It also enables optimization within the bounds of the learned prior [11, 16, 25] and weak-supervision when integrated in a differentiable form [27] into neural training processes [4, 20, 23, 42, 45, 65]. Closest to our approach in this category are the model fitting methods by [74] that textures and geometrically refines an untextured parametric quadruped model to zebra images and to [68] that uses optical flow to refine human pose. Although in a similar setting, our surface-free neural body model and volumetric rendering is fundamentally different to their textured triangle mesh that is rigged with forward kinematics and needs to be obtained a-priori.

**Implicit Body Models.**   A-NeRF bears close similarities with body models defined implicitly in terms of level-sets [58] and density of a sum of Gaussians [18, 48, 49] that are used for refining human pose, shape, and appearance via differentiable ray-tracing. However, sum of Gaussians and other primitives only provide rough approximations.

**Neural Scene Representations.**   Recent neural scene representations learn low-dimensional non-linear representations of meshes [28, 63], point clouds [1, 38, 67], sphere sets [24], and dense volumetric grids [29, 55]. Their respective geometric output representations enable rendering with classical rendering techniques but have limited expressiveness, e.g., due to the fixed connectivity of a surface mesh and large memory footprint of discretized volumes. This limitation is overcome by using unconstrained MLPs [39] paired with positional encoding [60] to characterize an arbitrary point in 3D space. Common are surface definitions via level-sets of the MLP that are rendered with sphere tracing [56] and density representations rendered with ray-marching [39]. The rendering step is required for maximum likelihood estimation, to define a likelihood over observable variables—real images—while learning a 3D model. The rendering can be learned too [40, 50, 51] but usually leads to inconsistencies, particularly when training data is scarce. Some concurrent works also use neural scene representations for refining camera motion [71], video reenactment [47], and facial models [12, 13]. Orthogonal to these works, our A-NeRF learns an articulated body model from estimated poses and uncalibrated cameras.

Closely related to ours is the NASA surface body model [10] that also defines an implicit function as the minimum of individual implicit functions that are rigidly attached to the bones of a skeleton, each conditioned on the entire human pose to model dependencies and learned from 3D scans. By contrast, we learn a volumetric model instead of a surface model and include appearance and rendering. Even more similar is the recent NeuralBody [46] representation, which combines a NeRF with a surface body model and underlying skeleton. By contrast to both approaches, we do not require surface supervision or initialization, condition pose differently, and refine pose, which enables us to learn from single videos in unconstrained environments.

## 3   Formulation

**Objective.**   Given a sequence $[\mathbf{I}_k]_{k=1}^{N}$ of N images $\mathbf{I}_k \in \mathbb{R}^{H \times W \times 3}$ stemming from one or several videos of the same person, our goal is to simultaneously estimate the time-varying skeleton poses $[\theta_k]_{k=1}^{N}$ and learn a detailed body model. Our A-NeRF body model $C_\phi$ is parametrized by neural network parameters $\phi$ that define volumetric shape and color while the skeleton captures motion over time. Figure 2 gives an overview of this generative model. It enables a rendering of the virtual body

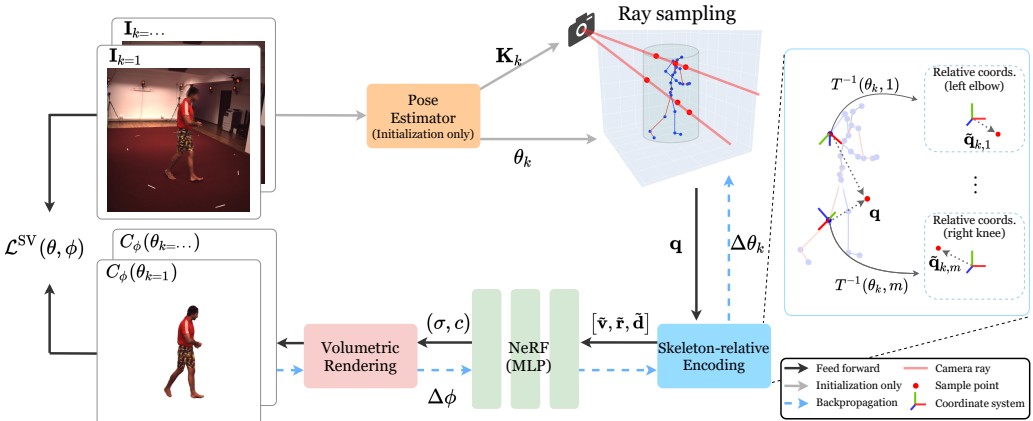

Figure 2: **Overview.** A-NeRF is a generative model that can be rendered and optimized on a photometric loss $\mathcal{L}^{\text{SV}}$ (white). First, the skeleton pose is initialized with an off-the-shelf estimator (orange). Second, this pose is refined via a skeleton-relative embedding (blue) that, when fed to NeRF (green), drives the implicit body model that is rendered by ray-marching (red). A key property of the skeleton-relative embedding is that a single 3D query location maps to an overcomplete reparametrization, with the same point represented relative to each skeleton bone (right).

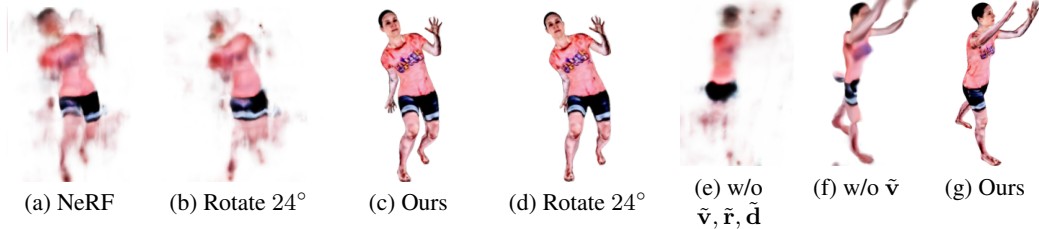

| (a) NeRF | (b) Rotate $24°$ | (c) Ours | (d) Rotate $24°$ | (e) w/o $\tilde{\mathbf{v}}, \tilde{\mathbf{r}}, \tilde{\mathbf{d}}$ | (f) w/o $\tilde{\mathbf{v}}$ | (g) Ours |

Figure 3: **Importance of our skeleton-relative encodings.** The original NeRF breaks (a) when training on a diverse set of poses and (b) further degrades when the poses are rotated. Even if (e) conditioned directly on $\theta_k$, the NeRF still suffers from artifacts due to the complexity and ambiguity of human articulation. With our skeleton-relative encoding (f, g), the geometry for the subject is consistent under rotation, and the quality is greatly improved, with the full model working best.

model in unseen poses and optimizes its parameters $\theta$ and $\phi$ on the image reconstruction objective

$$\mathcal{L}^{\text{SV}}(\theta, \phi) = \sum_k \underbrace{\|C_\phi(\theta_k) - \mathbf{I}_k\|_1}_{\text{data term}} + \underbrace{\lambda_\theta d(\theta_k - \hat{\theta}_k)}_{\text{pose regularizer}} + \underbrace{\lambda_t \left\| \frac{\partial^2 \theta_k}{\partial t^2} \right\|_2^2}_{\text{smoothness prior}}. \qquad (2)$$

with the influence of all three terms balanced by hyperparameters $\lambda_t$ and $\lambda_\theta$. The data term measures the distance between the images generated by $C_\phi$ and the input image with the L1 distance. The pose regularizer encourages the solution to stay close to an initial pose estimate $\hat{\theta}$ obtained from an off-the-shelf predictor [23], tolerating small shifts up to $\epsilon = 0.01$ with $d(x) = \min(\|x\|_2^2 - \epsilon, 0)$. Lastly, the smoothness prior penalizes acceleration $\frac{\partial^2 \theta_k}{\partial t^2}$ between poses of consecutive frames. Minimizing Eq. 2 can be seen as maximizing a corresponding probabilistic model, with the quadratic energy terms being the log-likelihoods of Gaussian distributions. Our focus is on formalizing the neural body model. For simplicity, we continue to write equations in terms of the objective functions used during inference with stochastic gradient descent.

## 3.1 NeRF and A-NeRF Image Formation Model

Instead of modeling the scene as a collection of triangles or other primitives, we define the human implicitly by a neural network as a function (Eq. 1) defined over all possible 3D points and view

directions in space [39]. Similar to NeRF, we render the image of the human subject via ray marching

$$C_\phi(u, v; \theta_k) = \sum_{i=1}^{Q} T_i(1 - \exp(-\sigma_i \delta_i))\mathbf{c}_i, \quad T_i = \exp\left(-\sum_{j=1}^{i-1} \sigma_j \delta_j\right), \tag{3}$$

with $(u, v)$ the 2D pixel location on the image, $i$ the index to 3D query positions $\mathbf{q}_i$ sampled along $\mathbf{d}$ and $\delta_i$ the distance to neighboring samples—a constant if samples would be taken at regular intervals. The $T_i$ is the accumulated transmittance for the ray traveling from the near plane to $\mathbf{q}_i$—the fraction of light reaching the sensor from sample point $i$. The $\mathbf{c}_i$ is the light color emitted or reflected at $i$. The final pixel color is the sum over all $Q$ samples, with the last sample taking the special role of the background. The background color is easily inferred as the median pixel color over the entire video for static camera setups. The ray direction $\mathbf{d} = \mathbf{K}_k^{-1}(u, v)$ is computed using the estimated camera intrinsics [23]. In the following, we introduce our skeleton parametrization $\theta_k$ and how to use it to effectively model dynamic articulated human motion.

### 3.2 Articulated Skeleton Pose Model

Our skeleton representation encodes the connectivity and static bone lengths via a rest pose of 3D joint locations. Dynamics are modeled with per-frame skeleton poses $\theta_k$, which define an affine transformation $T(\theta_k, m)$ for each bone $m$. Specifically, $T(\theta_k, m)$ maps a 3D position $\mathbf{p}_{k,m} \in \mathbb{R}^3$ in the $m$-th local bone coordinates to world coordinates $\mathbf{q} \in \mathbb{R}^3$ using homogeneous coordinates,

$$\begin{bmatrix} \mathbf{q} \\ 1 \end{bmatrix} = T(\theta_k, m) \begin{bmatrix} \mathbf{p}_{k,m} \\ 1 \end{bmatrix}, \tag{4}$$

where subscript $_{k,m}$ indicates that a variable is related to the $m$-th joint of image $\mathbf{I}_k$. Conversely, $T(\theta_k, m)^{-1}$ maps world to local bone coordinates. Note that our skeleton is equivalent to SMPL [30] and others, but without their parametric surface model, and can therefore be initialized with any skeleton pose estimator. We include more details of our skeleton representation in the supplementary.

### 3.3 A-NeRF Skeleton-Relative Encoding

Our core contribution is to transform the query locations $\mathbf{q}$ and view direction $\mathbf{d}$ relative to the skeleton before determining the color and opacity at that transformed point via NeRF. It is a form of reparameterization that explicitly incorporates domain knowledge of how the human body parts are linked and transformed relative to each other. Intuitively, our implicit formulation turns explicit models, such as SMPL [30], on its head. Instead of deforming the output surface via skinning, the query location is mapped in the inverse direction to local bone-relative coordinate 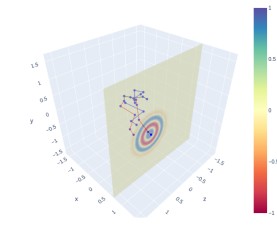 systems before processing through the NeRF network. Our final model uses a combined encoding $\mathbf{e}_k = [h(\tilde{\mathbf{v}}_k)\Gamma(\tilde{\mathbf{v}}_k), \tilde{\mathbf{r}}_k, h(\tilde{\mathbf{v}}_k)\Gamma(\mathbf{d}_k)]$ as input to the NeRF $F_\phi$. Note that the skeleton embedding introduces the desired time dependency, denoted by subscript $k$. The inlet shows our most crucial contribution, the relative distance encoding $\tilde{\mathbf{v}}_k$ followed by PE with **Cutoff** to reduce the influence of irrelevant bones. We derive the components of our encoding $\mathbf{e}_k$ and the other alternatives below.

- **Reference Pose Encoding** One could compensate motion by attaching the query $\mathbf{q}$ in world coordinates at frame $k$ to the closest bone $m$ and transforming it with

$$\mathbf{a}_k = T(\theta_0, m)T^{-1}(\theta_k, m)\mathbf{q}. \tag{5}$$

This puts the query relative to the bone $m$ as in frame $k$ but with the skeleton in rest pose $\theta_0$. NeRF could then learn without change in the 3D space of the rest pose as done before for surfaces [61]. However, this cannot capture non-rigid pose-dependent effects, such as muscle bulging, and has ambiguities when $\mathbf{q}$ is at equal distance to two bones.

- **Bone-relative Position (Rel. Pos.)** To remove these ambiguities and the ill-posed association to a single part, we map $\mathbf{q}$ relative to each bone $m$ with,

$$\tilde{\mathbf{q}}_k = [\tilde{\mathbf{q}}_{k,1}, \cdots, \tilde{\mathbf{q}}_{k,24}] \text{ and } \tilde{\mathbf{q}}_{k,m} = T^{-1}(\theta_k, m)\mathbf{q}. \tag{6}$$

The resulting individual bone coordinates are well suited to model the overwhelmingly rigid motion of the corresponding body part. Moreover, the overparameterization of position by concatenating all local encodings enables learning when complex interactions are necessary. However, such an embedding for all bones increases the dimensionality by an order of magnitude.

- **Relative Distance (Rel. Dist.)** Much simpler to compute are distances from $\mathbf{q}$ to all bones $m$,

$$\tilde{\mathbf{v}}_k = [\tilde{\mathbf{v}}_{k,1}, \cdots, \tilde{\mathbf{v}}_{k,24}], \text{ with } \tilde{\mathbf{v}}_{k,m} = \|\tilde{\mathbf{q}}_{k,m}\|_2 \in \mathbb{R}. \tag{7}$$

This radial encoding is used in our final model in favor of $\tilde{\mathbf{q}}$ because it naturally captures spherically shaped limbs, is lower-dimensional, and thereby improves reconstruction accuracy.

- **Relative Direction (Rel. Dir.)** Since the distance encoding is invariant to direction, we additionally obtain the direction vector to capture the orientation information of $\mathbf{q}$,

$$\tilde{\mathbf{r}}_k = [\tilde{\mathbf{r}}_{k,1}, \cdots, \tilde{\mathbf{r}}_{k,24}], \quad \tilde{\mathbf{r}}_{k,m} = \frac{\tilde{\mathbf{q}}_{k,m}}{\|\tilde{\mathbf{q}}_{k,m}\|_2} \in \mathbb{R}^3. \tag{8}$$

Note that by contrast to all other embeddings, this direction encoding did not profit from subsequent PE. We therefore pass it directly into $\mathbf{e}_k$.

- **Relative Ray Direction (Rel. Ray.)** NeRF models the illumination effects in a static 3D scene using the position and view direction. By contrast, our goal is to learn a body model that produces plausible colors with dynamic skeleton poses. Therefore, we transform $\mathbf{d}$ to obtain $\tilde{\mathbf{d}}$, the outgoing ray direction relative to each bone, similar to query position,

$$\tilde{\mathbf{d}}_k = [\tilde{\mathbf{d}}_{k,1}, \cdots, \tilde{\mathbf{d}}_{k,24}], \quad \tilde{\mathbf{d}}_{k,m} = [T^{-1}(\theta_k, m)]_{3\times3}\mathbf{d} \in \mathbb{R}^3, \tag{9}$$

with $[T^{-1}(\theta_k, m)]_{3\times3}$ the rotational part of the bone-to-world transformation $T^{-1}(\theta_k, m)$. Following concurrent works [33, 46], we also optimize an appearance code for each image to handle dynamic light effects. The combination of $\tilde{\mathbf{d}}$ and the per-image code enables A-NeRF to approximate the light effects in $\mathbf{I}_k$ plausibly. See the supplemental material for detailed discussions on modeling view-dependent effects in our setting.

- **Cutoff.** We desire a local embedding where points should not be influenced by all but only nearby bones. To this end, we introduce a windowed version of positional encoding by multiplying the encoding with respect to bone $m$ by $h(\tilde{\mathbf{v}}_{k,m}) = 1 - S(\tau(\tilde{\mathbf{v}}_{k,m} - t))$, with $S$ the sigmoid step function, $t$ the cutoff point, $\tau$ the sharpness. This leaves queries unaffected by distant bones.

Our embedding choice of $\mathbf{e}_k = [h(\tilde{\mathbf{v}}_k)\Gamma(\tilde{\mathbf{v}}_k), \tilde{\mathbf{r}}_k, h(\tilde{\mathbf{v}}_k)\Gamma(\mathbf{d}_k)]$ has the advantage of being invariant to the global shift and rotation of the person and preserves the piece-wise rigidity of articulated motion while still allowing for pose-dependent deformation (see Figure 3). In addition to $\mathbf{e}_k$, we also consider other embedding variants. See the supplementary and Section 4 for a detailed discussion.

## 4 Evaluation

We performed experiments to validate that A-NeRF learns accurate body models and poses, with fewer assumptions (single view, uncalibrated, and w/o a parametric surface model) than the related works. This makes it applicable to fine-grained pose refinement that improves the estimates of state-of-the-art methods. The supplements provide the implementation details, additional comparisons and ablation studies.

**Inference and Implementation Details** Our A-NeRF model is learned without supervision on a single or multiple videos of the same person. Camera intrinsics, bone lengths for setting $\mathbf{a}_m$, and pose $\theta_k$ are initialized with [23] for every frame $k$. These poses are then optimized on objective Eq.2, alongside the generative A-NeRF model. See supplementary for more details.

**Datasets.** We evaluate on the following benchmarks, and additionally on synthetic data created from **SURREAL** and **Mixamo** characters, which are listed in the supplement.

- **Human 3.6M [19]** The dataset consists of 5 training and 2 testing subjects (S9/S11) with ground truth 3D joint locations. We follow two widely adopted test protocols denoted as Protocol I [20, 22, 23] and Protocol II [41, 59], in which we evaluate 14/17-joint estimation error on every $5^{th}/64^{th}$ frame of the test videos, respectively. See the supplement for details.

- **MPI-INF-3DHP [35]** This dataset is a standard benchmark for human pose estimation. It consists of 4 indoor and 2 outdoor subjects with challenging human poses. The number of frames per subject range from 276 to 603.
- **MonoPerfCap [70]** The dataset consists of human performance video captured with a monocular camera in both indoor and outdoor settings. We use two subjects, Weipeng_outdoor and Nadia_outdoor, for our qualitative experiments. The two subjects have 1151 and 1635 frames, respectively, for training.

**Pose Metrics.**  We report the PA-MPJPE metric, the Euclidean distance between Procrustes-aligned (PA) predictions and ground truth 3D joint position averaged over all frames and joints of the test set. The PA alignment in scale and orientation is essential for comparing approaches that do not assume knowledge of the ground truth calibration and are, hence, ill-posed to the factors that the alignment removes. Following prior work [23, 22, 35], we also report percentange of correct keypoints (PCK) for MPI-INF-3DHP; the percentage of joints that lie within a distance of 150mm to the ground truth.

**Visual Metrics.**  We quantify the visual quality on MonoPerfCap and Human 3.6M datasets by training on a subset and testing on a held-out test set of the same character. Image quality is quantified via the PSNR and SSIM of the rendering compared with the reference image within the character bounding boxes. Because no ground truth pose is available in the required skeleton format on these datasets, we train our model once on the entire dataset to get reliable skeleton pose estimates as pseudo ground truth, and a second time with a part withheld to learn the body model for visual quality evaluation. Since MonoPerfCap has one sequence per actor, we exclude the last 20% of each video. For Human 3.6M we exclude entire actions, namely *Geeting-1,2*, *Posing-1,2* and *Walking-1,2*. The body model is then transferred to the held-out portion by using the pseudo ground truth poses as the driving motion. Thereby, we can still test the generalization of different models to new poses and viewpoints, irrespective of the underlying skeleton model provided in the dataset.

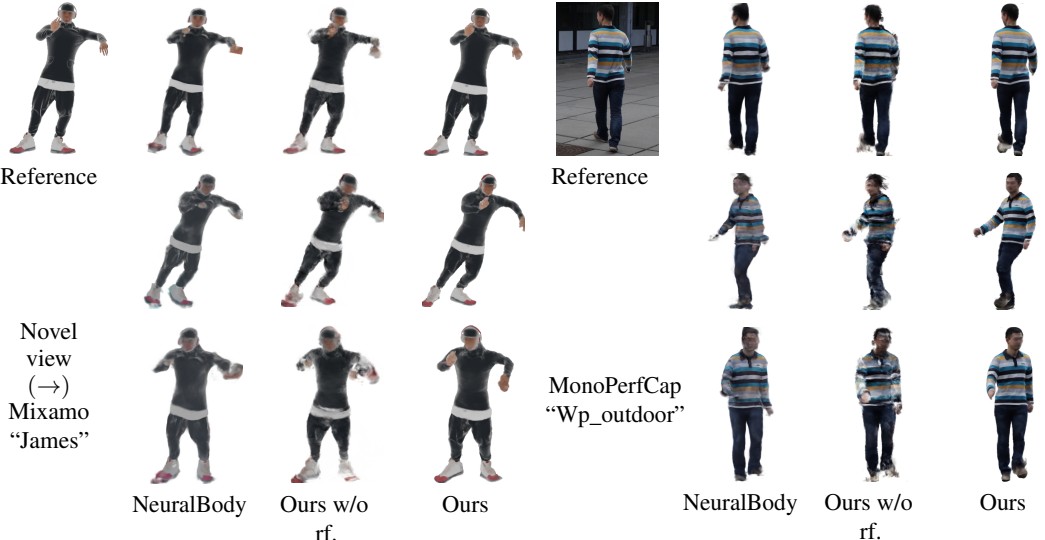

Figure 4: **Novel view synthesis from all models.** All models are trained with SPIN [23] estimated human pose and camera parameters. A-NeRF renderings (ours) align better with the reference images (top row), and the rendered novel views (2nd and 3rd rows) show better details (limbs, facial features).

**Novel-View-Synthesis and Character Animation.**  Our body model is generative, which allows us to train on a single or multiple uncalibrated videos and alter viewpoint and human pose. Figure 4 shows renderings of the same persons from a new camera angle; novel-view-synthesis. Likewise, Figure 5 demonstrates character animation, where the view is fixed, and the underlying skeleton is re-posed by manually changing joint angles or by transferring the motion between characters. A-NeRF is the first model that learns a detailed human body model with such capabilities without needing a 3D surface or multi-view supervision. While NeuralBody [46] can also learn photo-realistic human models from monocular images, their model anchors its representation on the SMPL 3D

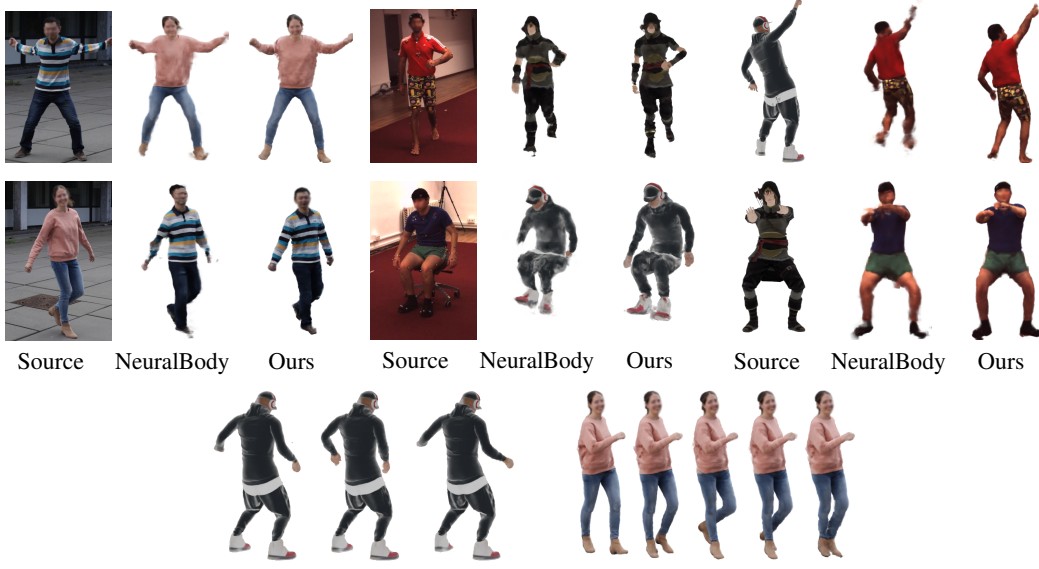

Source  NeuralBody  Ours  Source  NeuralBody  Ours  Source  NeuralBody  Ours

Figure 5: **Motion retargeting and animation. Top rows:** Pose transfer, with the source pose reconstructed by A-NeRF and rendered with different target body models. **Last row:** Animating the A-NeRF model while keeping the lower or upper part of the body fixed.

Table 1: **Quantitative evaluation on Human3.6M [19] and MPI-INF-3DHP [35]**. Our test-time pose refinement improves consistently upon the SPIN baseline, with largest improvements for extremities (PA-Wrist).

|  | Human 3.6M | | | | MPI-INF-3DHP | |
|  | Protocol I | Protocol II | Prot. II Wrist | Prot. II Multi-view | | |
| Method | PA-MPJPE↓ | PA-MPJPE↓ | PA-Wrist↓ | PA-MPJPE↓ | PA-MPJPE↓ | PCK↑ |
| MotioNet [53] | 54.6 | - | - | - | - | - |
| HoloPose [16] | 46.5 | - | - | - | - | - |
| VIBE [22] | 41.4 | n/a | n/a | n/a | **64.6** | **89.3** |
| SPIN [23] | **41.1** | - | - | - | 67.5 | 76.4 |
| Baseline (SPIN github [23]) | 42.7* | 41.9* | 66.5* | 34.0* | 68.2* | 79.3* |
| SPIN-SMPLify†[7, 23] | 57.7 | 59.2 | 100.9 | - | - | - |
| **Ours (w/o smoothness prior)** | 39.4 | **39.6** | **57.3** | **28.0** | 66.9 | 80.4 |
| **Ours** | **39.3** | n/a | n/a | n/a | **66.8** | **80.4** |

* Reevaluation of publicly-available model because missing evaluation protocols or not reproducible.

† We refine the SPIN estimated pose using SMPLify. We adopt the implementation from the SPIN repository.

surface. When trained on our single-view setup with the same noisy estimated poses $\hat{\theta}$ as input, NeuralBody suffers from artifacts and less detail when training, as it assumes 3D ground truth, which is only available in controlled conditions. The importance of our joint body and pose optimization is further validated by the ablation with refinement disabled (Ours w/o rf.), which similarly produces ghosting artifacts around extremities.

**Human Pose Estimation.** Training A-NeRF includes a form of test-time optimization (see Figure 1, right), only the initialization from [23] is trained supervised on the Human3.6M training set. Table 1 shows that on Human 3.6M, A-NeRF reaches comparable results with other single-view approaches, and achieves a $8.0\%$ improvement in PA-MPJPE upon the baseline used for pose initialization on Protocol I ($42.7 \rightarrow 39.3$) and $5.5\%$ on Protocol II ($41.9 \rightarrow 39.6$). Note that these are average numbers across all joints, including easy-to-predict hip, shoulder, and head joints. Our largest gains are on the extremities, e.g., with an improvement of $14\%$ (9.2 mm) for the wrist joint on Protocol II (PA-Wrist). We also compare to applying SMPLify [7], a method that refines 3D poses using 2D joint locations (estimated with [8]) as constraints, at test time. It tends to explain the 2D joint estimates perfectly but degrades the 3D pose. The final pose estimations become less accurate than the initial ones. In contrast, A-NeRF optimizes the poses by implicitly minimizing the disagreement

Table 2: **Visual quality evaluation on the Human3.6M [19] and MonoPerfCap [70] held-out sets**. Our full A-NeRF model significantly improves the visual quality. The body model itself attains a higher quality than NeuralBody (2nd vs. 3rd row), and additional detail is gained with the proposed pose refinement (last row).

| | MonoPerfCap | | Human 3.6M | |
|---|---|---|---|---|
| | PSNR ↑ | SSIM ↑ | PSNR ↑ | SSIM ↑ |
| NeuralBody, driving motion from [23] | 21.80 | 0.8476 | 22.08 | 0.8766 |
| NeuralBody, driving motion from A-NeRF refinement | 21.75 | 0.8468 | 22.55 | 0.8782 |
| A-NeRF w/o pose refinement | 21.99 | 0.8405 | 23.33 | 0.8776 |
| A-NeRF (Our full model) | **24.39** | **0.8851** | **27.45** | **0.9277** |

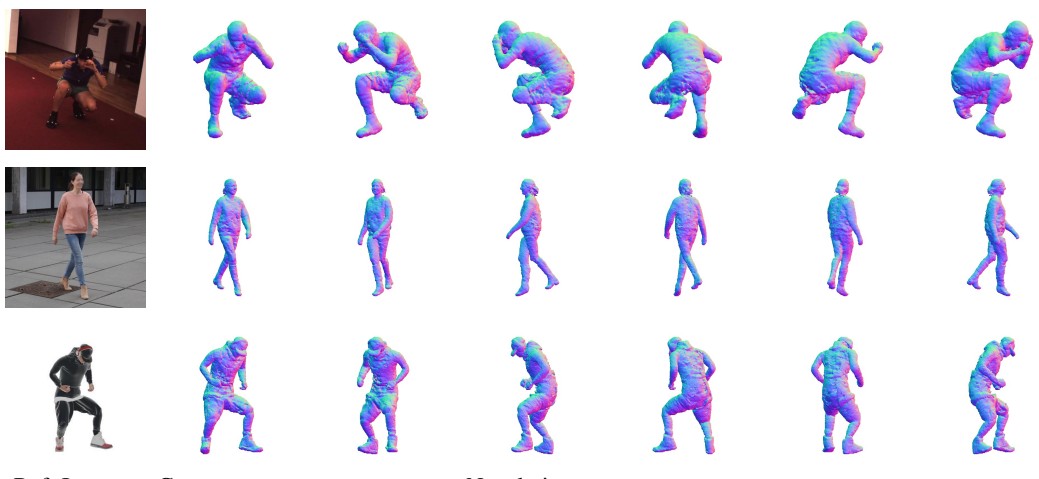

Ref. Image     Geometry →            Novel views

Figure 6: **Geometry.** The isosurface of our density model is rendered from multiple unseen views. A-NeRF learns plausible geometry without using explicit surface templates and accurate initial poses.

among the 3D body representation in different images, and thus achieves better performance. On MPI-INF-3DHP, Table 1 shows the results averaged over the 6 test subjects from MPI-INF-3DHP. Despite having a low number of frames and human poses available for learning the skeleton-relative encoding, A-NeRF still provides moderate improvements over the baseline estimations.

**Video-based volumetric reconstruction.** Figure 6 visualizes the learned density using Marching Cubes [31], with voxel grid resolution of 256 and density threshold 10. Despite only learned from monocular videos (no stereo, depth camera, or multi-view constraints) and without using a pre-defined template model, A-NeRF reconstructs a detailed volumetric body with details that could not be captured by offsets to a parametric surface model, such as the head phones and basebal cap in the last row of Figure 6. Note that no geometric smoothness term is enforced.

**Multi-view extension.** A-NeRF can also leverage multi-view refinement (Table 1, Protocol II Multi-view), even without access to ground truth camera calibration (see supplementary).

**Visual Quality Comparison.** We report the results in Table 2. We compare to NeuralBody with both the initial estimates from [23] and our refined pose since NeuralBody has no refinement step. Compared to both variants, our A-NeRF shows significantly better reconstruction performance on held-out poses. We observe that NeuralBody models can retain the facial features and hands as they anchor their representation on a 3D surface model. However, the rendered limbs and faces are blurry and distorted. Results are similar to training A-NeRF without refinement (see Figure 7). As the estimation for these joints is often inaccurate and noisy, the models without pose refinement simply learn to predict mean pixel values. To conclude, it is important to train with pose refinement, with which A-NeRF suffers less from artifacts with overall better visual quality.

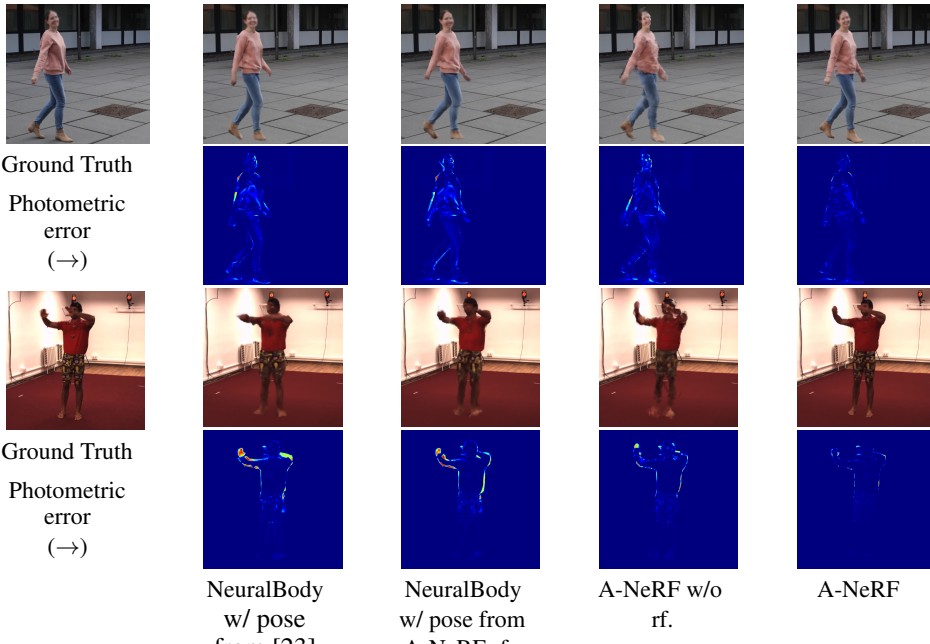

Ground Truth

Photometric error (→)

Ground Truth

Photometric error (→)

| NeuralBody w/ pose from [23] | NeuralBody w/ pose from A-NeRF rf. | A-NeRF w/o rf. | A-NeRF |

Figure 7: **A-NeRF with pose refinement generalizes better to test sequences.** We visualize both the rendered images, as well as the photometric error (squared distance, normalized to $[0, 1]$) between the rendered and the ground truth images (warmer color indicates higher error) from our MonoPerfCap (1-2th rows) and Human 3.6M (3-4th rows) held-out sets. NeuralBody models produce artifacts around body contours. A-NeRF w/o refinement cannot reproduce facial features and limbs. With pose refinement, A-NeRF can produce both appearances and shapes more plausibly.

**Ablation study.** Our detailed ablation studies are reported in the supplemental document. In summary, they reveal: i) Embedding relative 3D position, $\tilde{\mathbf{q}}_k$, instead of our proposed radial embedding $\tilde{\mathbf{v}}$ yields only half as good pose refinements. ii) Our embedding choices keep the dimensionality moderate while improving on or matching the PSNR and SSIM of higher-dimensional variants. iii) For a fixed number of images with accurate poses, learning from a long video with diverse poses has visual quality comparable to learning from multiple shorter clips.

**Limitations and Failure Cases.** Our computation time is the biggest bottleneck in extending A-NeRF to long sequences and multiple actors. Although a single static camera suffices, A-NeRF requires to see the person from all sides in varying poses to learn pose dependencies from data. The inlet shows a rendering of an extreme breakdance pose retargeted to a model trained on normal walking motions. Hence, the source pose is unseen during training and far from the data distribution, which leads to artifacts.

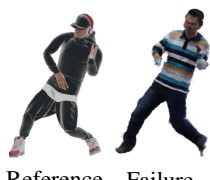

Reference    Failure

## 5    Conclusion

We propose a new way for integrating articulated skeleton models and implicit functions via an overcomplete re-parametrization. It includes learning an interpretable 3D representation from 2D images; a personalized volumetric density field with texture detail and time-varying poses of the actor depicted in the input. The underlying ill-posed problem of mapping a single query location to multiple parts is addressed with an overparametrization over nearby parts. To the best of our knowledge, A-NeRF is the first approach to define NeRF models for extreme and articulated motion on unconstrained video and this new approach scores high on the Human 3.6M benchmark. Importantly, it works from a single video and naturally extends to multi-view and does not require camera calibration in either scenario. This is an important step towards making motion capture more accurate and practical. In future work, we will learn a general human model from a database of subjects instead of individuals.

**Funding in direct support of this work:** NSERC Discovery grant, UBC Advanced Research Computing (ARC) GPU cluster, Compute Canada GPU servers, and a gift by Facebook Reality Labs.

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
