# OpenReview forum: "A-NeRF: Articulated Neural Radiance Fields for Learning Human Shape, Appearance, and Pose"
_NeurIPS.cc/2021/Conference — NeurIPS 2021 Poster_

### Official Review · Reviewer_gmac · 2021-07-07

**Rating:** 4
**Confidence:** 4

**Summary:**

This paper presents modelling articulated neural radiance fields for human dynamics from monocular videos. To model NeRF for dynamic human, it encodes 3d spatial points and ray directions w.r.t the articulated skeleton poses, which are fine-tuned together when training the NeRF field. Various skeleton-based feature encoding are experimented in consideration of both performance and memory.

**Limitations And Societal Impact:**

Seems ok.

**Main Review:**

There are 2 major contributions from the work. It first extends the static NeRF modeling to dynamic human motion by encoding the spatial points w.r.t. the bone transformations. Secondly, it utilizes the differential NeRF rendering to fine tune the estimated human pose that better aligns with image pixels. i am not fully convinced with either of them and have quite a few concerns (see detailed comments below).

- I do not find any sentences in the paper describing how the background is removed from the NeRF rendering. A foreground mask is needed for training NeuralBody and how do you obtain good masks for the evaluation datasets? Could the dark noise around the person come from imperfect segmentation? Figure A1 shows both your results and NeuralBody with a background and how do you obtain them? Related to this background question, is your image metrics (SSIM and PSNR) evaluated over the whole image or only for the region of interest (around the target human)? All of these important details are missing in the paper.

- The paper provides comparisons to NeuralBody and with ablation studies, demonstrates it is helpful to fine tune the estimated skeleton poses. I agree for in-the-wild videos, fine-tuning the poses seem a good idea. However,  it does not demonstrate the superiority of the proposed skeleton-based feature encoding over other approaches, such as anchoring over statistical human
surface as in NeuralBody. It seems to me that NeuralBody can also be easily extended to fine tune the skeleton but i realize it might be an unfair comparison to this submission. However, I would like to see both methods compared on synthetic dataset or dataset with ground-truth pose estimations for a better understanding on different feature encoding approaches.

- Pose estimation:  how important the regularization term is? The pose estimation error could theoretically be compensated by the NeRF field (overfitting). How do you ensure they are decoupled well? Besides, such a pose estimation procedure with NeRF training is very expensive and it does not seem a very fair comparison by comparing to feed-forward approaches (such as SPIN which the method is initialized from). I would suggest to also compare with some nonlinear optimization based methods, such as Simplify or Zanfir et al ECCV 2020, followed with SPIN initialization.

- Qualitative results:  it will be good to visualize the 3d geometry reconstructions from the trained NeRF field. The appearance looks very noisy for the subject in fig 1 and 3 (with clothing color diluted into the skin).  Any reasons?

- I think there could be a fundamental problem by transforming spatial points w.r.t. the skeleton as NeRF feature encodings. Suppose there is a constant lighting source in the scene and now how the subject is located w.r.t. this lighting source should largely affects its appearance. However, the proposed encoding scheme will result in global transformation invariant features and will the ANeRF produce right lighting effects, say the subject is rigidly translating in the scene?

- Table A7: for this monocular and multiview comparison, it is not clear from which camera views you evaluated the approach?

- Some missing related works:
   * D-NeRF: Neural Radiance Fields for Dynamic Scenes
   * Dynamic Neural Radiance Fields for Monocular 4D Facial Avatar Reconstruction


**Time Spent Reviewing:**

4

---

> ### Author Response · Authors · 2021-08-09
> **Response #R4 (gmac)**
>
> Thank you for your time and valuable feedback. Please check the shared comments above for general remarks.
>
> **How is the background removed from the NeRF rendering?**
>
> The model is rendered on white background by replacing the background sample in the ray-marching (line 139) with white. This visualization is used at test time to highlight a properly learned separation of foreground and background. No segmentation mask is used.
>
> We were unsure if this question also relates to the training. A-NeRF learns the foreground-background separation from an approximate background image that fills in any part not occupied by the human model. The background is obtained as the median pixel value over the whole image (lines 139-141). To improve efficiency, we sample rays from a mask obtained with DeepLabv3 (lines 235-236), which avoids sampling mostly from the background region. The mask needs not be precise and we dilate it by 5 pixels (Supp. 116-117) to not miss body parts.
>
> The same masks are used for training NeuralBody. In cases of extreme poses, the masks may be erroneous and not cover the entire body. These imperfect masks affect A-NeRF and NeuralBody in the same way. We now also tested a variant of NeuralBody with our background handling, to directly compare body model properties. Please see the answer in the next question (also, c.f. #R1 (EsYF) background handling).
>
> **Is the image quality evaluated on the whole image?**
>
> For evaluation in Table A1, we evaluate over tight bounding boxes (Supp lines 9-11) obtained by projecting the estimated SPIN poses to the image (same for both methods). This focuses the attention on the person while also including mispredictions close to the body. We fill in the background colors for both A-NeRF model and NeuralBody so that both predictions will not be penalized due to inaccuracy in the background UNLESS the models incorrectly put density there. Please see an explanation of the simpler background handling (tuned for studio settings) by NeuralBody in our answer to #R1 (EsYF) and how we now evaluate body model quality independent of the background handling. The improved background handling lessens floating artifacts but does not fix the slightly blurry and distorted limbs when retargeting to similar but unseen poses. Please see the new table added to #R1 (EsYF).
>
>
> **Is the skeleton-based feature encoding better than anchoring over the statistical human surface as in NeuralBody?**
>
> We believe our encoding and NeuralBody both have merits, we list the major ones in the general section. In particular, the following technical details are deciding:
> NeuralBody learns features attached to vertices of a predefined mesh that is controlled by shape and pose parameters. These features are further processed by a volumetric convolution step that distributes features on the surface into the neighboring space before running NeRF. This
> 1) allows it to model pose dependencies and shapes extending the surface model and
> 2) is efficient for rendering an entire image since the convolution provides features for neighboring NeRF samples at once.
> 3) embeds prior information on human body shape to learn from shorter videos/fewer examples.
>
> However, this choice has also drawbacks:
> 1) the volumetric convolution happens in pose-dependent object space (not on the un-posed mesh), which we speculate leads to the reduced generalization we observed.
> 2) the mesh posing and convolution, which are currently global operations, make evaluating individual rays inefficient.
>
>
> By contrast, A-NeRF
> 1) operates with an arguably simpler MLP that operates locally, thereby enabling sampling single rays from many images, which improves refinement and quality (c.f., old and updated tables A1 and A3)
> 2) provably generalizes better to new poses because of the skeleton-relative embedding (c.f. Figure 5, limb deformation)
> 3) does not require any body model and thereby applies to new domains and body shapes underrepresented in the current shape datasets.
> 4) additional steps necessary for pose refinement are introduced, such as the gradient accumulation and cut-off have not been explored by NeuralBody.
>
> **How important is pose regularization?**
>
> It helps but is not strictly essential. We will include the following numbers in Table A3.
>
> | Method       	| PA-MPJPE 	| Wrist Improve 	|
> |--------------|:----------:|:---------------:|
> | SPIN         	| 43.67    	| 0.00          	|
> | Ours w/o Reg 	| 41.21    	| 8.86          	|
> | Ours         	| 40.97    	| 9.02          	|
>
> We would also like to correct a typo in Table A3, first cell. The PA-MPJPE on S9 for SPIN is 43.67, the improvement column was correct.
>
> **Could mismatching pose theoretically be compensated by the NeRF overfitting?**
>
> A-NeRF has to represent the entire body in a compact form in its parameters. Overfitting by learning different shapes for every pose has high complexity. This 'bottleneck' thereby favours consistent models that generalize well. In principle, the model could drift. We believe that the distance-based encoding that is radial around bones imposes an implicit bias that keeps the bones in the center of the body (c.f. #R1 (EsYF) distance-based encoding). The remaining drift is further mitigated by the regularizer analyzed above.
>
> **Compare to Simplify or Zanfir et al ECCV 2020, initialized with SPIN**
>
> We added the suggested baseline: initialization with the same SPIN pose we use, followed by the Simplify fitting of the skeleton to 2D estimates from OpenPose under pose and shape priors. We used the SMPLify implementation included in SPIN. While this refinement improves a few frames, it worsens others when the 2D pose is off or ambiguous. In total, it decreases performance on H3.6M from 41.90 to 59.17 PA-MPJPE. This behaviour is probably the reason why SPIN did not report/advocate test-time optimization. In preliminary work, we already experimented with fitting SMPL to silhouettes, but maximizing image overlap often results in a higher 3D error since internal joints are not well constrained and clothing is not modeled in SMPL.
>
> Also, we would like to stress that our method does not only refine the poses, but also learns an articulated textured avatar during the process. This is different from SMPLify and Zanfir et al ECCV 2020. Additionally, it is also conceivable to integrate a 2D keypoint loss into A-NeRF, as in SMPLify, but this likely suffers from the same drawbacks.
>
>
> **The appearance looks very noisy for the subject in fig 1 and 3 (with clothing color diluted into the skin).**
>
> The appearance is virtually indistinguishable from the reference, see Supp. Video 00:57-01:36 for the ground truth. The artifacts stem from the low texture quality of the surreal dataset [F]. We will add reference images to the paper to clarify. The geometry is excellent for this and also of high quality for synthetic images (c.f. discussion with #R2 (2h5q)).
>
> [F] “Learning from Synthetic Humans”, Varol et al 2017.
>
> **A-NeRF produces the right lighting effects under translation?**
>
> A-NeRF produces plausible, but not necessarily physically correct light effects under rigid motion. Notice that, under translation, the relative ray encoding will change, which allows A-NeRF to learn position-dependent light effects should these occur in the training motion.
> See the discussion with #R3 (Zhik) about the plausible but non-physical generalization behaviour.
>
> **Which camera views are used for evaluating the approach?**
>
> All models in Table A7 are evaluated on the same test data, which has 5 novel views capturing 300 unseen poses. The testing views are obtained by rotating the training camera in both y-axis (by 32 degree) and x-axis (10 degree elevation in 4 views, and 22.5 degree in the 1 view).
>
> **Comparison to NeuralBody on ground truth poses and camera calibrations.**
>
> We added experiments that compare NeuralBody to A-NeRF in the synthetic setting used in Table A7 (400 pose case).  Precisely, both models are trained on Surreal with images from 3 out of 9 cameras in the training data, with 400 randomly sampled poses for 300K iterations. We similarly increase the batch size for NeuralBody so the number of training samples per batch will be the same (Supp. lines 128-131). Note that, we train only on 400 poses, as NeuralBody may not fit well on longer sequences (c.f. NeuralBody paper, Section 4.3). Both camera calibrations and training poses are perfect and error-free. Frame latents (Supp. lines 122-127) are disabled in both cases, for that there’s no per-frame-specific light effect.
>
> We evaluate both models on our test data to see how they generalize on both unseen poses and novel views (as in Table A7). The results are as the following:
>
> | Method     	| #pose 	| #imgs 	| PSNR  	| SSIM   	|
> |------------	|:------:|:-------:|:-------:|:--------:|
> | NeuralBody 	| 400   	| 1200  	| 23.86 	| 0.9304 	|
> | Ours       	| 400   	| 1200  	| **29.10** 	| **0.9655** 	|
>
> We observe that NeuralBody produces slightly deform/expanded body features for poses that are very different from the training data. We have contacted the authors of NeuralBody, and confirmed that such artifacts are expected for unseen test poses.

---

### Official Review · Reviewer_Zhik · 2021-07-12

**Rating:** 7
**Confidence:** 4

**Summary:**

This submission presents a way for adding articulation to neural radiance fields. The main contribution is an encoding of the query points and view directions relative to a skeleton. The proposed formulation is trained per subject on a video sequence and optimizes for the 3D skeleton configuration in each frame and a global MLP which represents the NeRF. This defines the motion tracking throughout the input sequence and the recovered model can be reposed by rendering frames with new joint locations.

**Limitations And Societal Impact:**

I think potential risks of creating models without consent of people is addressed. One point that could be emphasized more is that this method allows to render people in unseen poses which adds the risk of generating fakes.

**Main Review:**

Strengths:
- Adding an articulation model to a NeRF seems a novel and significant contribution
- Novel poses can be rendered photorealistically.

Weaknesses:
- The formulation seems very expensive to get the motion tracking of a single subject 60 hours of training on 2 high end GPUs is needed.
- It is unclear if the model can work with very short sequences. How long are the videos and how much pose variation in the videos is needed to avoid overfitting of the NeRF? My concern is that this might behave similar to when a vanilla NeRF is trained on sparse views and just overfit to the small number of poses.
- To me it is not clear how with the skeleton relative encoding the NeRF is still able to reason about view dependent lighting effects given the NeRF does not have access to the skeleton configuration. How is the network still able to recover the relative configuration of the lighting environment, view direction, surface position and orientation in the original world coordinate system. It seems without this it would be hard for the network to recover view dependent lighting effects.
- Similar to the comment above it is not clear to me how the network could understand how the lighting behaves under camera motion, given everything is expressed relative to the camera. From the pictures in the submission it is hard to tell if view dependent lighting effects are recovered. Example renderings that show how the appearance changes if the view direction input to the network is changed but the actual rendered view direction stays constant would be helpful to better understand the capabilities of the proposed network.


Summary:
May main concern with this paper is that the effect of the skeleton relative encoding on the view dependent effects is not discussed. Given that the view dependent effects is one of the key properties of NeRF I find it important that this would be discussed and evaluated better. At this point it is not clear to me if the proposed formulation is even still able to generate view dependent effects or if this part of the formulation could even be removed.


After Author Reply:
In the author's replies it turns out that the problem of modeling illumination more accurately is non-straight forward and the model proposed in this paper even though in my opinion slightly unintuitive works better than obvious alternatives. I don't think these issues need to be resolved for publishing this work. In my opinion the original submission is lacking a discussion of this limitation but I feel confident that this can be incorporated in the final version and left for future work. I adjusted my rating and time spent on the review accordingly.

**Time Spent Reviewing:**

3h

---

> ### Author Response · Authors · 2021-08-09
> **Response to #R3 (Zhik)**
>
> Thank you for your time and valuable feedback. Please check the shared comments above for general remarks.
>
> **High computational cost?**
>
> Yes, it is high (line 293). We hope to build upon speedups to NeRF in the future, such as concurrently developed hierarchical representations [A] and improved sampling strategies [B,C]. Furthermore, for reconstructing multiple videos of the same actor, the body model could be learned once and only pose be optimized for the others, which is significantly faster.
>
> [A] “KiloNeRF: Speeding up Neural Radiance Fields with Thousands of Tiny MLPs”, Reiser et al 2021
>
> [B] “AutoInt: Automatic Integration for Fast Neural Volume Rendering”, Lindel et al 2020
>
> [C] “NeRF in detail: Learning to sample for view synthesis”, Zisserman et al 2021
>
>
> **How much pose variation in the videos is needed to avoid overfitting of the NeRF?**
>
> For learning a body model with pose-dependent deformation from scratch it is indeed crucial to have sufficient pose variation. Otherwise, the model can overfit (c.f. discussion R1 (EsYF) on "why does it work"). In our experiments 1000 or more frames were sufficient (e.g. MonoPerfCap in Figure 5 is trained with 1151/1632 consecutive frames, and the Mixamo dataset with 1130 consecutive frames). Notably, A-NeRF can also learn from just 400 (non-consecutive) poses, given that the poses are diverse enough (as shown in Table A7). We will add results trained with few (<500), low-variance poses as failure cases, where appearance becomes view-dependent and geometry degrades to deformed blobs.
>
> Note that it is much easier to record one longer video with an uncalibrated camera than multiple, synchronized videos as required by most other methods.
>
>
> **View-dependent effects - without knowledge of the world coordinate system?**
>
> *Preamble* - Current NeRF implementations bake directional illumination into the scene encoding, which does not generalize to new illumination conditions or unseen views. We build upon the frame-dependent illumination codes proposed in [D,E,F] (Supp. lines 122-127, we will and should have mentioned this earlier in the main document) that allows us to model some illumination variation.
>
> *Discussion* - For approximating the rendering equation, the position in space, incoming light, and outcoming direction are required. It is natural to represent these quantities either in world or in object coordinates, relative to the surface; but the reference frame does not matter. In A-NeRF, we incorporate a (#frame x 16-dimensional) latent embedding, one 16-word for every frame, to handle frame-dependent illumination [D,E,F] (Supp. lines 122-127). This per frame embedding can learn to encode the illumination relative to the skeleton pose in that frame (we think of it as a compressed version of an environment cube map similar to how spherical harmonics work). Thereby, it encodes the incoming illumination (likely relative to a subset or a single reference bone, which is sufficient since a query point/ray is transformed to all bones). The bone-relative positional encoding defines the position in space (in each bone coordinate). The relative ray direction provides the outgoing direction. Having all quantities in bone-relative coordinates allows A-NeRF to explain view-dependent effects (see Supp. video 00:03-00:07 and Figure 4).
>
> *Experiments* - Triggered by the question on global orientation, we tried adding the ray direction in world coordinates (not skeleton-relative) to the encoding, which gives A-NeRF knowledge of the global orientation. We report the results on both image quality and pose accuracy below:
>
> --- Only on S9, Protocl II, PSNR/SSIM in the same setting as Table A1, PA-MPJPE as Table A3 ---
>
> | Ray Type              	| w/ frame latent 	| PSNR  	| SSIM   	| PA-MPJPE 	|
> |-----------------------|:-----------------:|:-------:|:--------:|:----------:|
> | None                  	| N               	| 25.89   	| 0.9132    	| 41.99    	|
> | None                  	| Y               	| 26.60   	| 0.9164    	| 41.54    	|
> | World Ray             	| N               	| 26.04   	| 0.9140    | 41.75    	|
> | World Ray             	| Y               	| 26.55 	| 0.9161 	| 41.41    	|
> | Rel. Ray. + World Ray 	| N               	| 26.25 	| 0.9170 	| 41.21    	|
> | Rel. Ray. + World Ray 	| Y               	| 26.64 	| 0.9192 	| 41.00    	|
> | Rel. Ray (ours)       	| N               	| 26.26   	| 0.9167    | 41.03    	|
> | Rel. Ray (ours full)  	| Y               	| 27.27 	| 0.9245 	| 40.97    	|
> (Y: has frame latent (Supp. lines 122-127))
>
> As shown above, learning with global orientation (World Ray) led to reduced performance and flickering results for novel view rendering (e.g., bullet-time effect). This indicates that the camera (world) causes overfitting in our single/estimated camera setting and that it is beneficial to encode all quantities relative to the skeleton. Not using any directional information (None) yields the worst results. We will add these variants to Tables A1 (image quality) and A3 (pose accuracy). See next Q&A for further discussion of why skeleton-relative encoding is beneficial and illumination computation is generally limited in our setting.
>
> [D] “Neural Body: Implicit Neural Representations with Structured Latent Codes for Novel View Synthesis of Dynamic Humans”, Peng et al 2021.
>
> [E] “Dynamic Neural Radiance Fields for Monocular 4D Facial Avatar Reconstruction”, Gafni et al 2021.
>
> [F] “NeRF in the Wild: Neural Radiance Fields for Unconstrained Photo Collections”, Martin-Brualla et al 2021.
>
>
>
> **View-dependent effects - illumination under camera motion?**
>
> A-NeRF models motion of the person well but is non-physical under camera motion. We assume a static camera, hence, there is no data point for our learning-based approach to generalize to camera motion. However, because all our quantities are encoded relative to the skeleton, rotating the camera (and light sources) or rotating the subject in the reverse direction are equivalent. Notably, that property is what lets us learn to predict novel views from human motion under a single camera. Hence, rotating the camera gives view-dependent effects that look plausible (as if the person would move relative to the camera and the light sources, see Supp. video 00:03-00:07, where the light appears to come from the top-left-frontal direction) but are not necessarily physically correct.
>
> In sum, A-NeRF can meaningfully learn view-dependent effects in the source sequence, and such complete modeling is important for accurate refinement (see next Q&A). Please note that we do not claim physically correct re-lighting. We will make this more explicit (lines 53-55). Furthermore, NeuralBody also uses frame codes and suffers from the same issues when applied to monocular or sparse viewpoints. Calibrating light sources from a single viewpoint is a research problem in itself and, to the best of our knowledge, has not been attempted for any NeRF model, yet, would be an interesting direction for future work.
>
>
> **View-dependent effects - could parts be removed?**
>
> Without relative ray direction (Rel. Ray.) encoding image quality degrades and pose refinement suffers drastically (None row in the above table). Other building blocks are ablated in the supplemental. Without any skeleton-relative embedding or similar reference structure, no correspondences could be made across frames; crucial for learning a body model that generalizes, as discussed above.

---

> > ### Comment · Reviewer_Zhik · 2021-08-26
> > **Representation of rendering equation**
> >
> > Thanks for the additional information about how the network is designed with the per frame illumination codes. Adding this to the main paper would be helpful for the understanding. While I agree that this would help to alleviate issues where the illumination is inconsistent between frames I do not see how this could help in approximating the rendering equation.
> >
> > In order to approximate the rendering equation the network would need to know the incoming light direction the viewing direction, geometry and material properties. As far as I can tell in the original NeRF due to the static scene the network would have access to all the information to learn to reason about all these quantities. Where I still disagree with the answer above is that the proposed architecture does have access to the incoming light direction. In my understanding of the submission the network would have no way to know about the incoming light direction.
> >
> > I do agree that experimentally this seems to lead to good numbers. However, the entry for None with frame latent seems missing from the table. Overall, I am a bit concerned about the fact that it is unclear why this model would work best and in the current manuscript this limitation does not seem to be discussed at all.

---

> > > ### Author Response · Authors · 2021-08-27
> > > **light representation in NeRF**
> > >
> > > Thanks for discussing these aspects further. (Due to personal constraints and the short time horizon, our response is not very polished)
> > >
> > > Our understanding of how illumination works in the original NeRF is as follows. Lights are not explicitly represented; there is no position of lights or other position or direction-dependent encoding of the (incoming) illumination. What we meant with 'bake directional illumination into the scene encoding' is that NeRF does not learn to reconstruct light positions and hence also does not reason about the incoming light direction. It only applies to a single frame, where the effects of incoming light from all directions are combined with the material properties to yield the outcoming light as a black-box function of the position and outgoing direction.
> > >
> > > The follow-up work [D,E,F] extends NeRF to represent changing illumination (in a static scene) with the frame codes. In their setting, the intensity and light position (e.g., sun) can change arbitrarily from frame to frame (no video continuity assumed). There is still no explicit modeling of the rendering equation/incoming light, but it is given the ability to overfit to changing illumination effects in every frame.
> > >
> > > In our case, the position of body parts and the illumination could change at the same time. When representing the illumination with black-box frame codes, it does not matter if the light positions changes, if the body changes, or both at the same time since they all change the effect of illumination and no individual factor (direction or change of light). In our first response, we argued that such a combined effect can equally well be learned in relative object coordinates as for the world coordinates in the original NeRF.
> > >
> > > We hence argue that our illumination handling makes as much/little sense as in the original. We agree that this is not the best way, but it is simple and was sufficient for our goals of learning a body model from scratch and refining the 3D pose. It would be valuable to model the interaction of material properties and illumination more explicitly in NeRF. It could be feasible in our setting to integrate existing approaches that use the human as a light probe and to exploit temporal smoothness of motion and illumination, perhaps also enforcing a constant illumination assumption. However, this would require going beyond radiance fields, to somehow model the reflectance properties instead of the radiance directly.
> > >
> > > **"None w/ frame codes"?** Due to the limited time, we were not able to produce all possible combinations of model components within the rebuttal period. Nevertheless, we expect the "None w/ frame codes" to be slightly better than "None w/o frame codes", similar to the distance between "World Ray w/o frame codes" and "World Ray w frame codes". Its performance is clearly bounded by these models that contain additional features and compare with and without codes. We did not test it because the other combinations clearly show that both aspects (outgoing ray direction and frame codes) are important. We will nevertheless try to get this result ASAP *(Edit: results added)*.

---

> > > > ### Author Response · Authors · 2021-08-27
> > > > **further thoughts**
> > > >
> > > > We should add that we generally concur with the reviewer's concern of how illumination is handled in dynamic NeRFs and appreciate the investigation and directions towards future work. We would be happy to include this discussion and associated experiments.
> > > >
> > > > Concerning the original question of whether the outgoing direction (Rel. Ray.) is helpful at all: In principle, our argument that the influence of the incoming direction is baked into the frame codes together with the overall illumination could be extended to include the outgoing direction too. If the incoming direction is implicit, why treat the outgoing direction explicitly? We believe that the difference is that the outgoing direction is known and its relative embedding varies under pose changes (thereby providing varied training data), while the incoming light is unknown. Including the known factors helps the network to learn their effect (as the experiments show). By contrast, the position of and direction to explicit light sources is unknown. Resorting to encoding the ray direction in world coordinates (**World Ray**) and letting the network **jointly** learn a representation of light sources and a function of how direction changes their effect did not succeed in our new experiments listed above. Disentangling these effects would require additional constraints (e.g., multiple views or assumptions on a sparse set of materials as used in related work for light source calibration and intrinsic image decomposition) or explicit rendering equations on reflectance, thereby superseding the existing NeRF formulations.
> > > >
> > > > As a variant to **World Ray**, we also started a new experiment (on top of our full model) in that we encode the three global coordinate axes relative to the skeleton, akin to the transformation of **Rel. Ray**. ~~It is at 1/5 of the total iterations and is on the same trajectory~~ *Edit: It performs the same (40.95 vs. 40.97 PA-MPJPE)* as our full model. In conclusion, none of the variants to more explicitly encode the effect of view-dependent illuminations helped. Therefore, we prefer our original model over more complex variants.

---

> > > > > ### Author Response · Authors · 2021-08-29
> > > > > **Table updated with "None w/ frame latent"**
> > > > >
> > > > > We have updated the table with **None w/ frame latent** to the table. As we claimed before, the frame codes bring similar performance gain as in the case of World Ray.

---

### Official Review · Reviewer_2h5q · 2021-07-13

**Rating:** 8
**Confidence:** 4

**Summary:**

The paper presents A-Nerf which is a generative neural body model capable of rendering a human under novel viewpoints and poses. A-Nerf is an extension of Nerf to capture dynamic human bodies. It's achieved by conditioning Nerf on the skeletal body pose information. The paper demonstrates that naively conditioning Nerf on the body pose information does not lead to optimal performance and therefore proposes a set of relative encodings of the pose information that results in significantly better renderings without the need of a full-body mesh model. The experiments are performed for motion capture as well as novel view synthesis where the proposed method demonstrates superior performance as compared to existing methods.

**Ethical Concerns:**

Ethical concerns have been addressed appropriately in the paper.

**Limitations And Societal Impact:**

Yes.

**Main Review:**

**Pros:**

+ The paper is well written and easy to read.
+ The proposed pose relative embeddings are novels and intuitive. I like the idea of windowed positional encodings to reduce the influence of parts that are far away from the sampled point.
+ The supplementary material provides an extensive ablation study of the proposed approach.
+ The proposed method provides better renderings as compared to the state-of-the-art method NeuralBody even though it doesn't require surface mesh.
+ I like the narrative of A-Nerf as a "Neural Body Model". Unlike existing body models, e.g., SMPL, A-Nerf can also capture additional finer details such as Caps, glasses, etc. that are otherwise not easily possible with existing human body models.
+ The paper demonstrates the ability to refine initial pose estimates by rendering pose-conditioned images using A-Nerf. This is a solid contribution of the paper. Unlike, other differentiable rendering-based methods, A-Nerf does not require a mesh model hence it can inspire future research into test-time optimization with model-free differentiable rendering.
+ The proposed approach doesn't require MV data to work and relies only on monocular video data.
+ The experiments are performed for motion capture as well as novel-view synthesis. The evaluation protocols are well-defined and reproducible.


**Cons:**

I cannot think of any significant limitation of the paper except its slightly incremental nature i.e., it mainly augments Nerf with pose conditioned embeddings.

**Questions to authors:**

- Have the authors tried learning the cutoff parameters?
- Have the authors visualized the underlying geometry of the person? Does it show anything reasonable?
-  Since most available methods for skeletal human pose estimation only provide joint positions, it will be great if the authors can comment on what will happen if the body pose information only consists of joints positions, rather than joint rotation angles and a T-pose? What has to be changed?


**Time Spent Reviewing:**

5

---

> ### Author Response · Authors · 2021-08-09
> **Response to #R1 (2h5q)**
>
> Thank you for your time and valuable feedback. Please check the shared comments above for general remarks.
>
> **Learning the cutoff?**
>
> We determined the cutoff parameter by a parameter search on the validation set (H3.6M S1). Learning bone-specific cutoffs is a promising direction for future work.
>
> **Geometric detail?**
>
> Computing a surface from our implicit density with marching cubes gives accurate results, similar to those reported in the NeuralBody paper. It reconstructs details such that faces are distinguishable, up to small-scale features such as the nose or feathers of the arrows in the Mixamo sequence. Only very fine details such as hair strands and the arrow shaft are missed. This additional experiment demonstrates that A-NeRF does not overfit but learns proper geometry. We will add these images to the main document and supplemental video.
>
> **Could the body model work on joint positions?**
>
> We started our derivation from a pure distance-to-joints encoding, but this indeed loses important directional information and led to mediocre results (c.f. #R1 (EsYF)  compared to radial encoding). It would be an interesting direction to explore if A-NeRF can start from positions and estimate the orientation from rough estimates, e.g., starting from an off-the-shelf skeleton fit to the 3D coordinates. It would likely require adding a strong 3D pose prior that models human joint limits as an additional regularization term.

---

### Official Review · Reviewer_EsYF · 2021-07-16

**Rating:** 6
**Confidence:** 5

**Summary:**

This paper aims to refine the 3D human poses. To this end, the authors equip neural radiance fields with human skeletons, where they design a skeleton-relative embedding based on the human skeleton. By training the A-NeRF on the video, the human poses are optimized to fit the observations.

**Ethics Review Area:**

["I don’t know"]

**Limitations And Societal Impact:**

I do not see potential negative societal impact of their work.

**Main Review:**

Originality:

1. This paper proposes a novel skeleton-relative embedding, which enables the neural radiance field to represent dynamic humans.
2. A-NeRF allows the human pose refinement during training, which improves the quality of rendering and estimated poses.

Quality:

1. Although the results in the submission look good, the proposed skeleton-relative embedding does not make sense to me. Why does relative distance work better than the bone-relative position. I think that reducing a position to a distance may cause ambiguity, since points on a spherical surface have the same distance.
2. For experiments, the results of NeuralBody always have black blobs, which seems strange. Is the NeuralBody trained with black background and visualized with the white background?
3. The training and test data is unclear. What are the training views and test views?
4. The motivation of this submission is unclear. At the first, I thought that it aims to refine the human poses with differentiable rendering. However, the experimental results are mostly about the view synthesis and animation. So I thought that it aims for view synthesis. However, the quantitative comparisons are only presented in the supplementary material, and the authors only compare their method with NeuralBody.

Clarity:

1. This submission is poorly written. Although I can understand the overall idea, it is still difficult for me to learn about the technical details. The authors could leave some unrelated contents in the supplementary material and describe their core method firstly.
2. For example, the readers probably cares about how to combine NeRF with human skeletons, so after describing the basic NeRF, the authors could introduce how to improve the input of NeRF based on 3D skeletons. The position encoding (line 163), 4D space-time encoding (line 169), and conditioning on pose (line 172) are unrelated with the skeleton-relative embedding and should be put in other places.
3. Similarly, the NeRF and articulated skeleton pose model are well-known techniques, which can be described more simply and can be put in the supplementary material.
4. The quantitative comparisons on view synthesis and the ablation studies are important, which should be put in the experiment section, while not in the supplementary material.

Significance:

1. This paper develops a human representation for view synthesis and animation from sparse camera views. It probably enlighten follow-up works in the field of human view synthesis.

Typos:

1. Line 201: "Much simpler to compute are distances from q to all bones m".

**Time Spent Reviewing:**

4 hours

---

> ### Author Response · Authors · 2021-08-09
> **Response to #R1 (EsYF)**
>
> Thank you for your time and valuable feedback. Please check the shared comments above for general remarks.
>
> **Why does relative distance work better than the bone-relative position? Loss of information?**
>
> Note that we encode *relative distance (Rel. Dist)*  and *relative bone direction (Rel. Dir.)* relative to the bone, a form of polar coordinates that has no information loss. We believe this radial encoding adds an implicit bias that helps to model humans (highest image quality in Table A5) because limbs are generally rounded with the bone close to the center. Moreover,  it lets us reduce dimensionality by tuning positional encoding resolutions for distance and orientation separately to maximize visual quality (Table A5).
>
> **Quantitative image quality comparisons only in supplemental?**
>
> We will complement the qualitative results in the main document by moving Table A1 from the supplemental while compactifying the background section as suggested.
>
> **Background handling, NeuralBody trained with black background but visualized on white?**
>
> NeuralBody was designed to be trained in studio conditions with a monochrome background replaced to black. To apply it to real images, we segmented the human using DeepLabv3 (line 234-236 as for our method). Note that our approach is not learned with white background either but an approximated background color (Line 139-141). The visualization on white background hence reveals artifacts for both methods, unless the background in the image is white.
>
> To provide a fairer comparison on the NeuralBody models, excluding background handling, we now modified NeuralBody with our background (our BG) handling and re-evaluated Table A1:
>
> | Method                             	| MonoPerfCap (PSNR) 	| (SSIM) 	| Human 3.6M (PSNR) 	| (SSIM) 	|
> |------------------------------------|:--------------------:|:--------:|:-------------------:|:--------:|
> | NeuralBody                         	|        19.68       	| 0.8141 	|       20.38       	| 0.8250 	|
> | NeuralBody, refined motion         	|        19.73       	| 0.8137 	|       20.40       	| 0.8255 	|
> | NeuraBody + our BG                 	|        21.80       	| 0.8476 	|       22.08       	| 0.8766 	|
> | NeuraBody + our BG, refined motion 	|        21.75       	| 0.8468 	|       22.55       	| 0.8727 	|
> | A-NeRF w/o refinement              	|        21.99       	| 0.8405 	|       23.33       	| 0.8776 	|
> | A-NeRF                             	|        24.99       	| 0.8851 	|       27.45       	| 0.9277 	|
>
> We will also update corresponding figures. We arrive at the same conclusion as our original claim in Supp. lines 24-27: NeuralBody + OurBG retains better facial features and body structure than A-NeRF w/o refinement; the improved background handling lessens floating artifacts but does not fix the slightly blurry and distorted limbs when retargeting to similar but unseen poses.
>
> In particular, we will replace the 5th column in Figure 4 and the 2nd column in Figure 3, with NeuralBody trained with white BG since the current replacement to black background is an unfair disadvantage on the already white background. We apologize for not explaining this shortcoming of the comparison when using the author's implementation without further tuning.
>
> **Training and test data?**
>
> We will explicitly label test views and quantify the difference to the training:
>
>
> - Figure 4, Figure A2, and Supp. video (00:23-00:33), the novel viewpoints are generated by regular sampling from two different camera trajectories: (1) cameras spaced from 0 to 360 degrees (bullet-time effect), and (2) a circular trajectory that rotates from -25 to 25 degrees along y-axis, -15 to 15 degrees along x-axis (elevation), with additional zoom-in and out effect.
> - Figure A1 and Table A1, report on novel poses, from the 20% of unseen poses excluded from the training (Supp. lines 10-16).
> - Table 1 and A2-A3 on H3.6M and MPI-INF-3DHP are test-time optimizations of 3D pose (transductive learning setting).
> - Inlet at line 293, shows a novel view and pose from retargeting (stemming from an entirely different actor).
> - For Table A4-A7, the training is a dome of 9 different cameras (to compare single and multi-view reconstruction), five at 0 degree elevation, two at 30 degree, and two at 45 degree. The 5 test cameras are at 10 and 22.5 elevation and rotated along y (in relation to the training views) by 32 degrees.
> We include the exact values in our code repository to aid reproduction and future comparisons.
>
> **Pose refinement is the main goal but most evaluation on image quality. Comparisons beyond NeuralBody?**
>
> Our goal is to learn neural avatars from in-the-wild data, and pose refinement is indeed a core component for this task (see Figure 4, Figure 5, and Table A1). Therefore our evaluation is two-fold:
>
> 1. We directly compare against the most accurate skeleton pose estimation approaches (Table 1). This also indirectly compares to all the other approaches reporting on the established H3.6M and MPI-INF-3DHP datasets. We also showcase the gained improvements visually (see Figure 4, Figure 5, Figure A1, Supp. video 00:24-00:32). We will add side-by-side comparisons to the main competitors. In addition, we added one more comparison to applying SMPLify/SPIN at test time (c.f. #R3 (Zhik) on other iterative methods).
>
> 2. Improved image quality is important for pose refinement, as it comes with and explains the improved pose accuracy. It also has merit in itself, demonstrating that an accurate body model is learned that can generalize to new views and motion retargeting applications. (c.f. #R3 (Zhik) for further discussion).
>
> **Only compared to NeuralBody for image quality?**
>
> NeuralBody showed to outperform other, surface-based neural rendering models and was hence chosen as the closest and strongest competitor. The dynamic NeRF models suggested by R4 (gmac) do not apply to large articulated motion. Please note that we also compare against a range of other baselines, using simpler skeleton-relative embeddings.
>
> **Reorganization**
>
> We will compactify the background section to create more room for ablations and qualitative results that are currently presented in the supplemental material. We found it difficult to judge how well the relatively recent NeRF is known in the community. Thanks for the calibration.

---

> > ### Comment · Reviewer_EsYF · 2021-09-01
> > **Response**
> >
> > Thanks for your detailed responses!
> >
> > Most of my concerns are resolved, so I would increase my rating to acceptance.
> >
> > Please reorganize the paper as you promise.

---

### Author Response · Authors · 2021-08-09
**Shared Comments**

### General remarks

We thank the reviewers for their time and very constructive reviews. We will answer open questions for every reviewer below. In the following, we give an overview of two recurring topics and link to the individual discussions.

**Is the skeleton-relative embedding sound and necessary?**

We demonstrate in the existing ablation study and an additional experiment that all model components are important and justified, including the distance-based embedding and approximation of view-dependent effects. Please refer to the discussion for #R1 (EsYF) for the encoding benefits and #R3 (Zhik), #R4 (gmac) on the reasoning for simplifying assumptions on directional illumination in the monocular setting.


**Is A-Nerf better or worse than NeuralBody?**

Both have their merits and are largely orthogonal:

Our A-NeRF
1. learns body shape and appearance from scratch instead of using surface models constructed from hundreds of laser-scans, by skeleton-relative embedding (a promising basis for non-human domains),
2. enables learning from uncalibrated cameras w/o GT 3D pose, for which refinement and collecting information from distant frames is absolutely crucial (NeuralBody struggles with long videos and does not provide refinement, see their paper Section 4.3),
3. provides improved generalization to novel poses, which enables retargeting applications (NeuralBody yields wobbly artifacts when retargeted)

NeuralBody
1. requires fewer frames for training by embedding features on a well-defined surface
2. the sparse volumetric convolution step and not doing pose refinement improves runtime,
3. on synthetic sequences both are comparable, NeuralBody is better on short sequences A-NeRF on longer ones.

Please see R4 (gmac)'s Q&A on refining pose with NeuralBody for additional discussion.

---

### Decision · Program_Chairs · 2021-09-27

**Decision:**

Accept (Poster)

**Comment:**

The paper proposes A-Nerf, a generative neural body model capable of rendering a human under novel viewpoints and poses. A-Nerf is an extension of Nerf to capture dynamic human bodies. This is achieved by conditioning Nerf on the skeletal body pose information. The paper demonstrates that naively conditioning Nerf on the body pose information does not lead to optimal performance and therefore proposes a set of relative encodings of the pose information that results in significantly better renderings without the need of a full-body mesh model.

Reviewers raised concerns regarding the similarity or advantages to the proposed formulation with NeuralBody, which the rebuttal addressed to some extent. Reviewers agree on the novelty of the formulation, specifically the spatial encodings for ray and query point with respect to the skeleton bones. The paper is recommended for publication.